# Latent Back-projection Network: a missing-wedge generative model for cryo-electron tomography

## Abstract

Cryo-electron tomography (Cryo-ET) is hindered by the missing wedge, a gap in Fourier space information caused by limited tilt series angular coverage, leading to anisotropic resolution loss and artifacts. We introduce a Latent Back-projection Network (LBN) which learns a latent representation of the imaged volume, enabling re-projecion of micrographs at tilt angles which cannot be experimentally collected. Our model is trained as a masked autoencoder on a large dataset of real Cryo-ET tilt series. This allows the model to learn to realistically impute missing information, with the appropriate physical constraints imposed by our back-projection and re-projection architecture. Through evaluations on real tilt series with excluded tilts, the missing wedge of simulated tilt series, and synthetic augmentation to tomogram reconstruction, LBN demonstrates generative capabilities which overcome limitations of prior approaches. Additionally, LBN is designed as a general model, providing a path toward an easily distributed pre-trained method for missing-wedge correction.

## 1 Introduction

Cryo-electron tomography (cryo-ET) is a powerful imaging technique that enables the three-dimensional visualization of biological specimens, such as macromolecular complexes and cellular structures, in their native state (Nogales & Sjors, 2015) (**Figure 1A**). This is achieved by acquiring a tilt-series, comprising 2D projections obtained by rotating the sample in an electron microscope at various angles, typically in 1–3° increments (Young & Villa, 2023). These projections are then computationally combined using tomographic reconstruction algorithms, such as weighted back-projection (WBP), to generate a 3D tomogram of the imaged volume. However, a critical limitation in this process is the missing wedge, a region of uncollected data in the Fourier space caused by restricted tilt angles, typically limited to $\pm 60°$ due to mechanical constraints, radiation damage, and sample warping (Radermacher, 2007) (**Figure 1B**).

The missing wedge introduces anisotropic resolution, elongation artifacts, and loss of structural details, significantly hindering the interpretability of reconstructed tomograms. All annotation, segmentation, or subtomogram averaging approaches applied to the tomogram to produce useful structural information are downstream of this initial reconstruction process which is affected by the missing wedge. All reconstruction algorithms are limited in the quality of tomogram they can produce by the input tilt series they receive. Therefore, we propose a method which intervenes in this process at the level of the tilt series.

Although information is absolutely absent in Fourier space due to the missing wedge, the requirement that missing tilt angles remain physically consistent with observed tilts constrains the possible reconstructions. We approach missing wedge correction as a general problem: given the set of constraints imposed by observed tilts, we seek to generate the most probable missing tilt.

The Latent Back-projection Network implements these constraints through its architecture, an image-formation-aware variant of an auto-encoder. By back-projecting tilt latents to

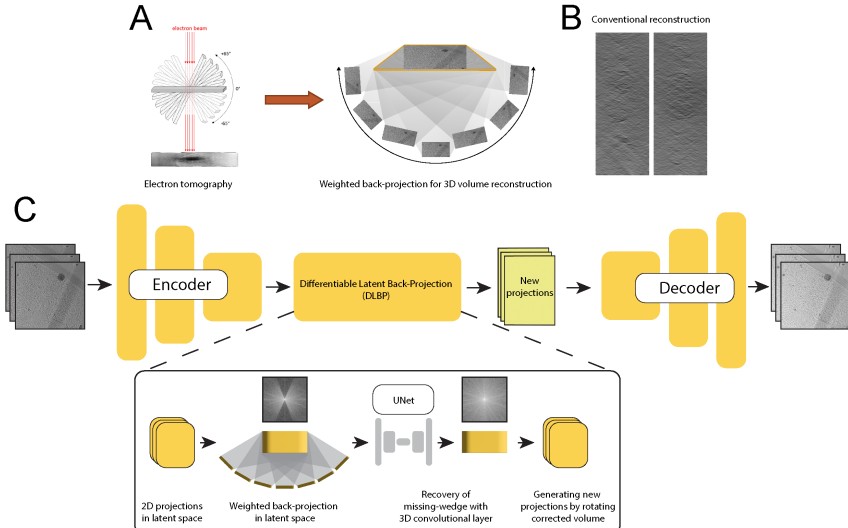

Figure 1: Illustration of tomographic reconstruction and missing wedge: **(A)** Illustration showing acquisition of projections on Cryo-electron microscope and conventional back-projection reconstruction to obtain 3D volume. **(B)** Representative images showing a ZX slice throw the tomogram volume illustrating resolution degradation due to the missing wedge. **(C)** Detail Latent Back-projection Model.

reconstruct a volume latent and then correcting and re-projecting that volume we enforce a physical consistency between generated tilts. Self-supervised training through tilt-masking allows the model to learn to useful corrections to the volume latent.

This intuitive approach to tilt synthesis outperforms both the baseline of classical back-projection and re-projection as well as an existing deep learning missing-wedge correction approach, IsoNet (Liu et al., 2022). This improvement in performance results in both masked-tilt evaluations on collected tilt-series as well as direct missing-wedge evaluations on simulated tilt-series. We additionally demonstrate the use of synthesized tilts to generate an improved reconstruction.

The contributions of this paper are as follows:

- Introduce image-formation-aware auto-encoder architecture for Cryo-ET and self-supervised training approach

- Achieve superior performance over baselines and prior approaches on recovering missing tilts in real and simulated datasets

- Demonstrate downstream use of synthesized tilts for improved tomographic reconstruction.

## 2 RELATED WORK

Cryo-ET enables high-resolution 3D imaging of biological structures but is limited by the missing wedge, a gap in Fourier-space data due to restricted tilt-series angular coverage, typically ±60°, causing artifacts and anisotropic resolution, specifically, reduced resolution in the depth dimension (Radermacher, 2007). Traditional reconstruction methods, such as filtered back-projection (FBP, another name for WBP) (Rit et al., 2016) and iterative techniques like simultaneous iterative reconstruction (SIRT) (Yan et al., 2019), process tilt-series projections but exacerbate missing wedge artifacts, failing to recover high-frequency details (Radermacher, 2007). Computational approaches, such as total variation minimization (Cai et al., 2022) and Fourier space interpolation (Ková čik et al., 2014), aim to reduce artifacts but often over-smooth features or introduce non-physical results.

Deep learning has introduced promising solutions for cryo-ET challenges. Generative adversarial networks (GANs) have been particularly impactful for tasks like denoising, reconstruction, and data simulation. In single-particle cryo-EM, CryoGAN (Gupta et al., 2021) uses adversarial learning to reconstruct 3D structures from 2D projections without pose estimation, optimizing a 3D model to match real projection distributions. While effective, CryoGAN focuses on single-particle analysis and does not address the missing wedge in cryo-ET. In cryo-ET, CryoETGAN (Wu et al., 2022) uses a cycle-consistent GAN to generate synthetic tilt-series images, helping to increase data for tasks such as classification but not directly correcting the missing wedge. Our approach is also capable of interpolating between observed tilts to increase data, but here we focus on it's ability to correct for the missing wedge. These GAN-based methods often require large datasets and careful tuning to avoid artifacts, limiting their generalizability.

Several deep learning approaches have targeted sinogram "in-painting" to fill missing projections in electron tomography. For instance, Ding et al. (2019) proposed a GAN with Residual-in-Residual Dense Blocks to "in-paint" missing wedge sinograms, followed by a UNet GAN to reduce artifacts in reconstructed tomograms. More recently, UsiNet (Yao et al., 2024) introduced an unsupervised U-Net for sinogram inpainting in nanoparticle electron tomography, effectively correcting the missing wedge without ground truth data. While these methods show promise, they either rely on adversarial training, which can be unstable, or are tailored to non-biological samples, differing from cryo-ET's complex biological structures.

Cryo-DRGN (Zhong et al., 2021) is a notable deep learning method in cryo-EM, using a variational autoencoder to model conformational heterogeneity by learning a latent space of 2D projections. While powerful for capturing structural variability in single-particle cryo-EM, Cryo-DRGN does not address the missing wedge in cryo-ET, as its focus is on molecular dynamics rather than projection synthesis for reconstruction.

IsoNet (Liu et al., 2022) is designed for cryo-ET, employing convolutional neural networks to in-paint missing frequencies in reconstructed tomograms to restore isotropic resolution. However, IsoNet operates on pre-reconstructed tomograms, which have already lost information due to the missing wedge. Potentially limiting its ability to recover high-resolution details and often producing over-smoothed results (Wu et al., 2022). In contrast, methods that leverage original tilt-series projections, which retain richer information, offer a more direct approach to missing wedge correction.

Our Latent Back-projection Network (LBN) introduces a novel framework by directly synthesizing new tilt-series projections in the frequency domain, bypassing traditional tomogram reconstruction. Unlike GAN-based methods like CryoGAN and CryoETGAN, LBN avoids adversarial training challenges by using a differentiable back-projection mechanism guided by the Fourier slice theorem, ensuring physical consistency. While Cryo-DRGN models structural heterogeneity and IsoNet processes degraded tomograms, LBN leverages the rich information in tilt-series projections to generate projections at unseen angles, preserving high-frequency details. Compared to sinogram inpainting methods like Ding et al. (2019) and UsiNet, LBN's latent space encoding and 3D back-projection offer enhanced generalization across diverse biological samples. No other approach combines LBN's projection-based synthesis, frequency-domain processing, and differentiable back-projection, making it uniquely suited for high-resolution cryo-ET. Trained on public datasets, LBN outperforms conventional and state-of-the-art methods, advancing structural biology for radiation-sensitive specimens. By synthesizing tilts in the missing wedge, the upsampled tilt series can then be input into any tomographic reconstruction algorithm, providing an approach that can be easily integrated into existing Cryo-ET pipelines.

## 3 Method

The overall workflow is divided into three stages (**Figure 1C**).

1. Each 2D projection ($P_n$) is first encoded into a compact latent representation ($L_n$), which helps us efficiently perform back-projection. We input a set of $N$ tilts, each projections of the same sample at distinct angles, $\theta_n$ for $n = 1, \ldots, N$. For this,

we utilize a ResNet-style encoder with down-sampling power of 8x, $E_\phi(\cdot)$ (with $\phi$ representing trainable parameters).

2. Next, the latent projections $\{L_n\}_{n=1}^N$ are processed with our Differentiable Latent Back Projection (DLBP) module, using the associated angles $\theta_n$ to back-project all projections into a single 3D latent volume $V$. To mitigate the effects of the missing wedge, which arises due to the limited range of projection angles, the back-projected volume $V$ is processed by a small UNet-style network ($UNet_\phi(\cdot)$) designed to learn and recover the missing information in the volume.

3. To generate new projections at arbitrary angles, the 3D latent volume $V$ is manipulated. Specifically, for a new angle $\theta_m$, the volume is rotated and a new latent 2D projection is obtained by summing over the $Z$ dimension. This simulates the physical projection operation in tomography, where integrating along a direction yields a 2D projection. Finally, the latent projection $L_m$ is decoded back to the original resolution ($P_m$) using a ResNet-style decoder, $D_\phi(\cdot)$.

This process is detailed in Appendix A.2. Note that the filter used ($filter(\cdot)$) does not have any learned parameters, but is instead a ramp filter which emerges from the formulation of weighted back-projection, as shown in Appendix A.3. We compute the rotations of the talents in a differentiable manner by defining affine matrices for the given angles and performing grid sampling. Additionally, the pseudo-code frames the encoding and decoding steps as sequential solely to make explicit the independence of the number of input projections and the number of synthesized projection but, in practice, these steps are performed in parallel.

The ResNet-like architecture of the autoencoder consists of double convolution blocks with residual connections, with the latent being 8x downsampled in the spatial dimensions compared to the input tilts. The 3D UNet which is applied to the latent volume includes an additional 4x downsampling. This component of the architecture allows the model to learn to correct artifacts in the latent reconstruction performed by back-projection. This model is designed to address the particularities of representing cryo-EM data. As micrographs are typically large images with a high SNR, we opt to perform reconstruction in the latent space, as the learned compression should be able to reduce noise and the smaller spatial dimensions reduce the computational cost of reconstruction. The use of back-projection and re-projection in our architecture induces constraints on the synthesized tilts which preserve the correct physical relationship between tilts. This results in the major benefit of our approach to missing-wedge correction: the generated tilts must be physically consistent with the observed tilts. The details of the architecture are given in Appendix A.2.

### 3.1 Training the model

To train the Latent Backprojection Network (LBN) to recover missing wedge information, we employ a masked tilt scheme. In each iteration, up to 80% of the tilt projections for each tomogram are randomly masked, and the LBN is tasked with reconstructing both observed and missing tilts. To prevent overfitting to the fixed missing wedge orientation, we apply data augmentation by rotating the tilts by a random angle offset in the range $[-90°, 90°]$. The model is trained for 2200 epochs using mean squared error (MSE) loss, a learning rate of $1e^{-5}$, and the Adam optimizer, with learning rate reduction on validation loss plateaus.

We also implement an auxiliary loss based on the formulation of Wasserstein GANs. As our auto-encoder model does not model the latent space as a random variable, we instead measure the ability to differentiate between the model input and output reconstructions. In this sense, the critic network, trained along side our model, represents more of a custom perceptual similarity loss than an actual GAN. The critic is applied patch-wise, as a convolutional network that outputs a map of scores for an input, with the loss utilizing the mean difference in scores (or mean score). The auto-encoder and critic network are trained in an alternating manner, either maximizing difference in the critic's score between the input tilt and the output of the autoencoder or simultaneously minimizing the MSE and maximizing the critic

score for the output of the autoencoder. With the critic as $C()$, the critic loss is given by

$$L_C = \frac{1}{n} \sum_{i=1}^{n} C(P_i) - C(D(E(P_i))) \tag{1}$$

The autoencoder loss is then given by:

$$L_A E = \frac{1}{n} \sum_{i=1}^{n} (P_i - D(E(P_i)))^2 - C(D(E(P_i))) \tag{2}$$

## 4 RESULTS

### 4.1 EXPERIMENTAL SETUP

To evaluate the LBN for recovering the missing wedge in cryo-ET, we trained the model on approximately 730 publicly available tilt-series datasets, encompassing diverse resolutions, biological content, and tilt angle ranges. Specifically, we collected dataset with ID from 10000 to 10048 from the CZI cryoET Data Portal (2024), containing ribosomes, membranes, microtubules, etc. with varying resolutions, and tilt angle ranges ($\pm40$ - $\pm60$) (specific dataset IDs listed in A.1). We then held-out tilt series from DS-10000, DS-10004, DS-10009, DS-10010, and DS-10016 to use as a test set. The remaining tilt-series were used as training data. Performance is evaluated on a held-out test dataset, enabling quantitative comparisons with conventional and state-of-the-art methods.

### 4.2 RECOVERING MASKED TILTS

First we compare the LBN against IsoNet, a state-of-the-art method, and conventional reconstruction (e.g., weighted back-projection) on the held-out validation dataset. For IsoNet and the conventional method, which output 3D tomograms, we re-project volumes at ground-truth tilt angles. We compare these and LBN synthesized projections to ground-truth tilts by MSE and Pearson correlation. We compare all methods using a leave-1-tilt-out experimental scheme.

Conventional reconstruction methods generate tomograms omitting one tilt angle at a time, and IsoNet (pre-trained on 5 randomly selected training dataset) was used to correct them. We then re-project these at the omitted angle and compare with the ground truth. The LBN predicts each missing tilt projection using all other available/remaining tilts.

Table 1: Model comparison on test dataset.

| | Mean over all angles | | Mean correlation for selected angle range | | | | |
| --- | --- | --- | --- | --- | --- | --- | --- |
| Model | MSE | Correlation | < -20° | > -20° | 0° | < 20° | > 20° |
| Conventional reconstruction | 1.02 | 0.15 | 0.13 | 0.15 | 0.22 | 0.17 | 0.18 |
| IsoNet | 1.16 | 0.23 | 0.18 | 0.23 | 0.28 | 0.25 | 0.26 |
| **LBN** | **0.63** | **0.57** | **0.58** | **0.53** | **0.53** | **0.55** | **0.59** |

We quantitatively compare the LBN against IsoNet and conventional reconstruction on the hold-out validation dataset, with results summarized in **Table 1**.

### 4.3 GENERATING MISSING WEDGE

As ground truth projections in the missing wedge cannot actually be collected on an electron microscope, the only possible way to directly evaluate the ability to synthesize tilts in the missing wedge is via simulated data. For the sake of a reproducible evaluation, we used the ground truth volumes from SHREC 2021 as the source of simulated tilt series. We computed projections from these volumes with the same parameters as originally done for SHREC 2021, except for intervals of one degree in between $-89°$ and $89°$. We provide our model and a WBP reconstruction algorithm with all tilts between $-66°$ and $66°$. and re-project at the collected range.

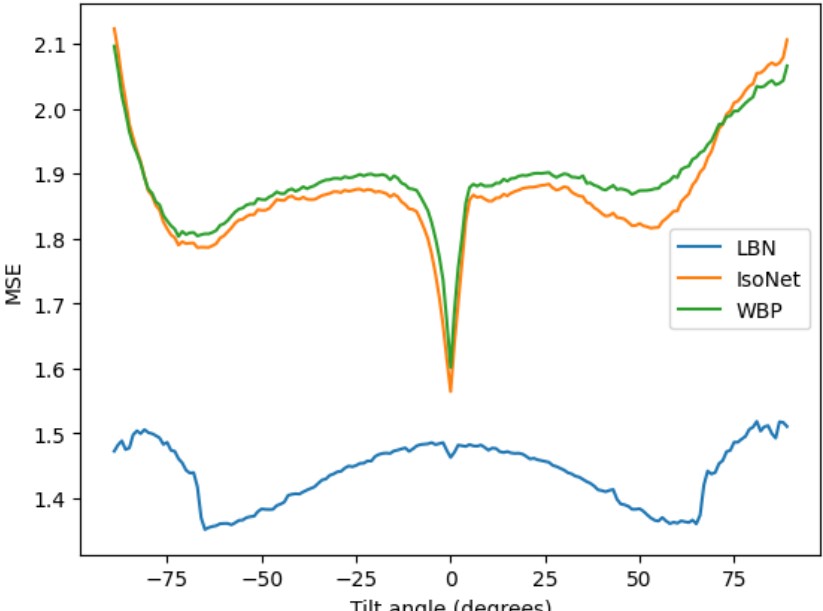

(a) Line plots showing LBN, IsoNet and WBP performance on recovering missing tilts from [leave-1-out] experiment with

(b) Visualization of re-projected tilts in leave-one-out evaluation, depicting DS-10004 Position 516.

Figure 2: Quantitative and qualitative results of leave-one-out task

Figure 3: Line plots showing LBN, IsoNet and conventional reconstruction performance on recovering missing-wedge tilts.

## 4.4 Utilize synthesized tilts in reconstruction

In order to directly evaluate the ability of the model to upsample the tilt series in the missing-wedge, we again utilize the simulated tilt series produced from SHREC 2021. In this case, we also reconstruct a tomogram using the entire projected tilt range. We compute the Fourier Shell Correlation (FSC) between this full-information reconstruction and a

Table 2: Model comparison on simulated dataset.

| | Mean over all angles | | Mean mse for selected angle range | | | | |
|---|---|---|---|---|---|---|---|
| Model | MSE | Correlation | < -66° | > -66° | 0° | < 66° | > 66° |
| Conventional reconstruction | 1.889 | 0.06 | 1.888 | 1.862 | 1.60 | 1.882 | 2.005 |
| IsoNet | 1.869 | 0.07 | 1.886 | 1.837 | 1.564 | 1.848 | 2.018 |
| **LBN** | **1.43** | **0.29** | **1.477** | **1.425** | **1.46** | **1.427** | **1.484** |

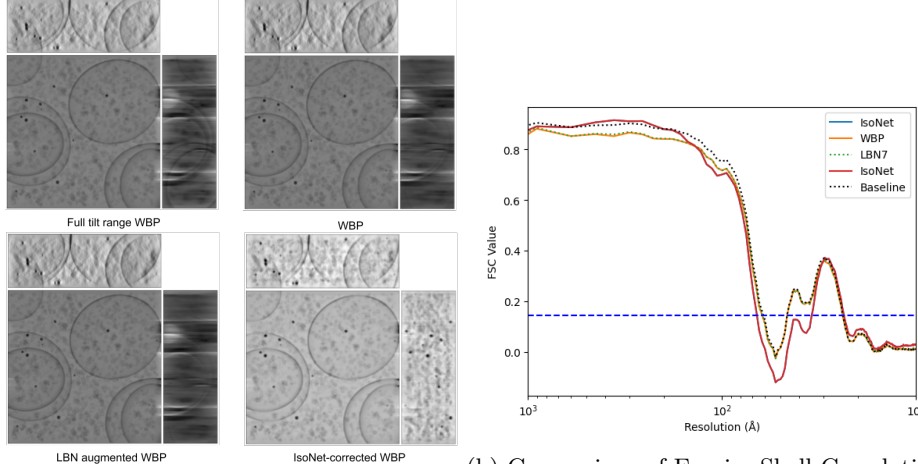

(a) Orthogonal projections of the reconstructions of model 0 of SHREC

(b) Comparison of Fourier Shell Correlation (FSC) for LBN, IsoNet, and WBP for model 0 of SHREC

Figure 4: Results on reconstructing model 0 of SHREC using synthesized tilts, in comparison to IsoNet, WBP with the full tilt range, and WBP with the missing wedge.

reconstruction using a tilt-series augmented with LBN-generated tilts at all missing-wedge angles. We also compute the FSC with the missing-wedge reconstruction as a baseline and the FSC with the IsoNet-corrected tomogram.

As shown in Figure 4b, we do not see a clear improvement in the effective resolution (as determined by the intersection of the FSC curve with the 1/7 threshold) achieved by synthesizing missing tilts using the LBN. However, IsoNet also does not appear to improve the affective resolution. It is worth noting that FSC resolution here is a global measure, while the missing wedge problem primarily impacts effective resolution along the z-axis. It is possible that the overall noise is obscuring nontrivial differences in the effective resolution along the z-axis between these models.

The FSC results align with the qualitative results of the augmented reconstruction, as shown in Figure 4a. The gaps in the membrane reveal missing wedge artifacts present in all reconstructions.

## 4.5 ABLATION STUDY

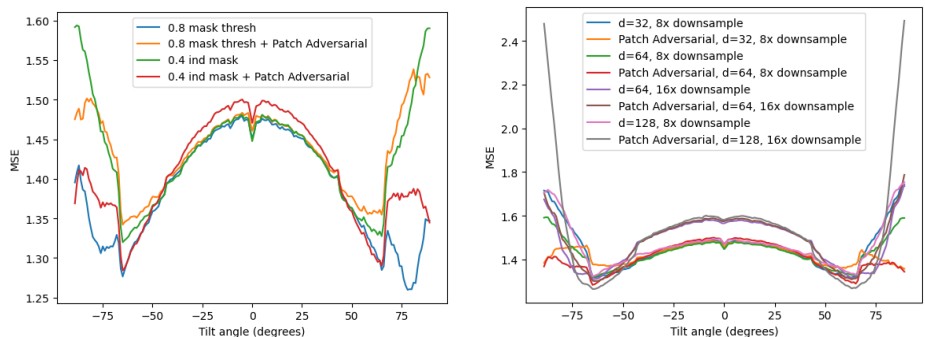

(a) Ablation of masking strategy during training

(b) Ablation of model architecture, d is size of latent dimension

Figure 5: Different variants of the LBN models performance on the missing wedge generation task using SHREC

To study the contribution of different components of the model, we evaluated several variants in an ablation study. The first design component which we varied was the masking scheme utilized during the training of the network. The first approach to masking tilts is to set a threshold for the maximum proportion of tilts to mask (here 0.8) and then uniformly randomly sample a number between 0 and that threshold. We then randomly (uniformly) sample tilts to mask to reach the sampled masking proportion. The alternative approach we studied is to independently randomly mask each tilt with a given probability (here 0.4) and resampling if all tilts are masked. This approach offers less control over the amount of masking, but avoids linking the probability of masking different tilts together.

The second design component ablated is the patch adversarial loss. This is a major variation in the design as it involves a second network trained in an alternating manner with the LBN model. The patch adversarial classifier augments the loss to help the reconstruction model address pereceptual features that are not well captured by mean square error.

Our ablation study uses models trained for 500 epochs instead of the 2200 epochs for our original experiment. As visible in Figure 5a, the thresholded masking approach with uniform sampling appears most effective without the adversarial loss at 500 epochs, however we have found that this approaches falls short with longer training. The 0.4 masking with the adversarial loss is the other approach that demonstrates the capacity to generate missing tilts after 500 epochs and this approach remains stable with longer training, which is why it is the approach utilized in the prior shown experiments.

Additionally, we ablated the size of the latent dimension, $d$, (the number of channels, not the spatial dimension, which is dependent on input size). We also did effectively ablate the size of spatial dimensions of the latent as well through showing results for a model which utilizes 16x downsampling (from input to latent) rather than the original 8x downsampling. These results, shown in Figure 5b, demonstrate that increased downsampling introduces a considerable gap in the ability to reconstruct observed tilts. Finally, a smaller latent dimension noticeably impedes the ability to generalize to unobserved tilts.

## 5 Conclusion

In all evaluations, our model surpasses both the baseline of WBP and the only previously published deep learning method for missing-wedge correction, IsoNet. Our intuitive latent back-projection architecture is able to learn to generate not collected tilts which are correctly physically constrained by the collected tilts. This general purpose model will be able to broadly improve the output of Cryo-ET initial reconstructions, which could lead to further improvements in downstream tasks like subtomogram average or annotation.

As shown in Figure 4, LBN results in an extremely slight improvement in local resolution over the baseline. However, in comparison to IsoNet, which exhibits a very small drop in local resolution in comparison to the baseline, this represents a success.

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

## A APPENDIX

### A.1 LIST OF PUBLIC DATASETS USED DURING TRAINING

Within the presented work, we used the dataset hosted at CZI cryoET Data Portal (2024) with CZI IDs ranging from 10000 to 10048, some of which have also corresponding EMPIAR or EMDB identifiers:

EMPIAR IDs: EMPIAR-10988, EMPIAR-10988, EMPIAR-10989, EMPIAR-10731, EMPIAR-10499, EMPIAR-10493, EMPIAR-11658, EMPIAR-11058, EMPIAR-11078, EMPIAR-11370, EMD-27654,

and

EMBD ID: EMD-17241, EMD-17242, EMD-17243, EMD-17244, EMD-17245, EMD-29922, EMD-29923, EMD-29925, EMD-29924, EMD-29922, EMD-29923, EMD-29925, EMD-29924, EMD-29922, EMD-29923, EMD-29925, EMD-29924, EMD-29922, EMD-29923, EMD-29925, EMD-29924, EMD-29922, EMD-29923, EMD-29925, EMD-29924, EMD-29922, EMD-29923, EMD-29925, EMD-29924

## A.2 Implementation Details

All models were trained on a single NVIDIA A100 GPU with 80GB VRAM. Each model required 50-70 GB of VRAM during training depending on the downsampling used (x8 or x16) and the size of the input data. Each model required approximately 1.5 GPU days to train. 3 additional variants of the model were designed during the model development process. The models were implemented in PyTorch and all code utilized was original.

The autoencoder model utilized 5 blocks of double convolutions in the encoder, with the second convolution always having a 3x3 kernel and stride 1 and the first convolution with the following (kernel size, stride) pattern: [(11,2), (7,1), (7,2), (3,2), (3,1)]. Every convolution was followed by a GroupNorm operation with [1, 4, 4, 8, 8] groups. The decoder mirrored this structure. The 3D UNet used on the latent volume consisted of 4 double convolution blocks in the encoder, all with 3x3 kernels and the second and 4th block having stride 2 for their second convolutions, again with a mirrored decoder. The UNet utilized GroupNorm with 8 groups. In all models, LeakyReLU with a negative slope of 0.5 was utilized as an activated function following each convolution. The critic model is a 5-layer residual convolutional neural network, with the first layer being a 7x7 convolution and the remaining layers being 3x3 convolutions. The critic has 6r output channels and utilizes ReLU activations and GroupNorm with 4 groups after each convolution is applied.

---

**Algorithm 1** Latent Back-Projection

---

$P_n \in \mathbb{R}^{C \times Y \times X}$
$\theta_n \in \mathbb{R}; \; -\pi/2 < \theta_n \leq \pi/2$
$\theta_m \in \mathbb{R}; \; -\pi/2 < \theta_m \leq \pi/2$
$E_\phi(P_n) \to L_n$ : ResNet-like Encoder
$D_\phi(L_n) \to P_n$ : ResNet-like Decoder
**for** $n = 1, \ldots, N$ **do**
    $L_n \leftarrow E_\phi(P_n)$
    $L_n \leftarrow IFFT(filter(X_n) \cdot FFT(L_n))$
    $rot_n \leftarrow compute\_affine\_matrix(\theta_n)$
    $L_n \leftarrow grid\_sample(L_n, rot_n)$
**end for**
$V \leftarrow \sum_n L_n$
$V \leftarrow UNet_\phi(V)$
**for** $m = 1, \ldots, M$ **do**
    $rot_m \leftarrow compute\_affine\_matrix(\theta_m)$
    $V_m \leftarrow grid\_sample(V, rot_m)$
    $L_m \leftarrow project(V_m)$
    $P_m \leftarrow D_\phi(L_m)$
**end for**

---

## A.3 Derivation of Weighted Back-projection

The Fourier Slice Theorem states that a projection of a volume taken at angle $\theta$ is equal to a slice of the Fourier space representation of that volume at angle $\theta$. Therefore, the conceptually simplest approach to tomographic reconstruction is to take the Fourier transform of each projection, rotate it the angle it was recorded at, combine all projections in the Fourier domain and then compute the Inverse Fourier Transform. Following the example of Kak & Slaney (2001), we demonstrate that Weighted Back-projection is equivalent to this process.

Let a projection at angle $\theta$ (with respect to the x-axis) of a volume be given by

$$P_\theta(q, r) = \int_{-\infty}^{\infty} f(q, r, s) ds$$

where $q = x\cos(\theta) + z\sin(\theta)$, $r = y$, and $s = -x\sin(\theta) + z\cos(\theta)$.

Then, it's Fourier transform is given by

$$S_\theta(t, v) = \int_{-\infty}^{\infty} \int_{-\infty}^{\infty} P_\theta(q, r) e^{-j2\pi(tq+vr)} dq \, dr$$

Now, examining the inverse Fourier transform of the volume, we can derive the equation for weighted back-projection.

$$f(x, y, z) = \int_{-\infty}^{\infty} \int_{-\infty}^{\infty} \int_{-\infty}^{\infty} F(u, v, w) e^{j2\pi(ux+vy+wz)} du \, dv \, dw$$

$$u = t\cos(\theta)$$

$$w = t\sin(\theta)$$

$$du \, dw = t \, dt \, d\theta$$

$$f(x, y, z) = \int_0^{2\pi} \int_0^{\infty} \int_{-\infty}^{\infty} F(t, v, \theta) e^{j2\pi(tx\cos(\theta)+vy+tz\sin(\theta))} t \, dv \, dt \, d\theta$$

$$= \int_0^{\pi} \int_0^{\infty} \int_{-\infty}^{\infty} F(t, v, \theta) e^{j2\pi(tx\cos(\theta)+vy+tz\sin(\theta))} t \, dv \, dt \, d\theta$$

$$+ \int_0^{\pi} \int_0^{\infty} \int_{-\infty}^{\infty} F(t, v, \theta + \pi) e^{j2\pi(tx\cos(\theta+\pi)+vy+tz\sin(\theta+\pi))} t \, dv \, dt \, d\theta$$

$$F(v, t, \theta + \pi) = F(-v, t, \theta)$$

$$f(x, y, z) = \int_0^{\pi} \left[ \int_{-\infty}^{\infty} \int_{-\infty}^{\infty} F(t, v, \theta) |t| e^{j2\pi(tx\cos(\theta)+vy+tz\sin(\theta))} dv \, dt \right] d\theta$$

$$= \int_0^{\pi} \left[ \int_{-\infty}^{\infty} \int_{-\infty}^{\infty} S_\theta(t, v) |t| e^{j2\pi(tx\cos(\theta)+vy+tz\sin(\theta))} dv \, dt \right] d\theta$$

$$= \int_0^{\pi} Q_\theta(x\cos(\theta) + z\sin(\theta), y) d\theta$$

$$Q_\theta(q, r) = \int_{-\infty}^{\infty} \int_{-\infty}^{\infty} S_\theta(t, v) |t| e^{j2\pi(tq+vr))} dv \, dt$$

The equation we derive for weighted back-projection,

$$f(x, y, z) = \int_0^{\pi} Q_\theta(x\cos(\theta) + z\sin(\theta), y) d\theta$$

is an integral over only the projection angles, thus, through the change of variables, we are able to combine the projection in the space domain rather than the Fourier domain, which simplifies interpolation.

As shown above, $Q_\theta(q, r)$ is essentially a filtered version of the projection $P_\theta(q, r)$ where filter is applied by weighting the frequencies using $|t|$ in the Fourier domain. These filtered projections are then back-projected to create the final reconstruction. In practice, for discrete implementations of this algorithm, zeroing out the zero frequency term results in a problematic loss of information, so the actual filter used is derived from the impulse response of a band-limited version of $|t|$ (the frequency band being set by the Nyquist frequency).

### A.4 Ablation Study

In the case of leave-one-out reconstruction, the adversarial loss results in a relatively minor difference in final performance. The independent masking scheme does seem to be slightly

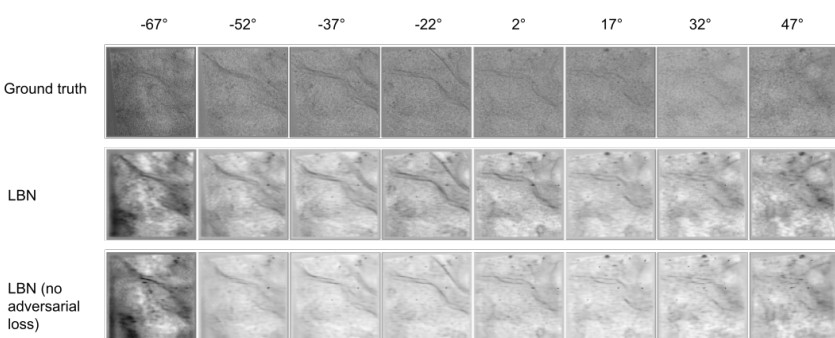

Figure 6: Visualization of re-projected tilts in leave-one-out evaluation, depicting DS-10004 Position 516

more performant than the uniform masking scheme, and the patch adversarial loss seems to offer an improvement for the uniform masking scheme but not across the board. Although the improvement from the patch adversarial loss appears to be very minor in terms of MSE, there is a qualitative improvement in contrast and clarity of the re-projected tilts, as shown in Figure 6.

