# OpenReview forum: "Latent Back-projection Network: a missing-wedge generative model for cryo-electron tomography"
_ICLR.cc/2026/Conference — Submitted to ICLR 2026_

### Official Review · Reviewer_LM1V · 2025-10-22

**Soundness:** 2
**Presentation:** 1
**Contribution:** 2
**Rating:** 2
**Confidence:** 4

**Summary:**

The authors propose a special network architecture to reconstruct Cryo-ET images and correct for the missing wedge in the measurement setup. They test this network on a dataset and report better results.

**Strengths:**

- The problem of recovering images under constrained measurement setup (i.e., where some part can not be measured) is important. Not only in Cryo-ET, but also for example in limited angle computed tomography. It would be interesting to see if
 the network architecture can potentially be adapted to other reconstruction tasks or if it limited to Cryo-ET

**Weaknesses:**

Overall, the paper presents a specific network architecture for one specific reconstruction task. The experiments are not really exhaustive and do not include several important ablations.

- There are no images of the reconstructed volumes. It would be useful to also compare the visual quality (or the performance of downstream applications)
- There are no ablations regarding architecture (depth of the encoder, dimension of the latent space, etc.), training setup or training dataset
- There is no ablations regarding the additional auxiliary loss (line 235). Is this loss necessary?
- The description of the method is a bit short (not even one page)
- Formatting:There is a large image before the title, this should be moved somewhere else in my opinion

**Questions:**

- It is a unclear to me how the back-projection interacts with the latent space? I guess the back-projection is based on the physical geometry (i.e, tilt angles etc.), but the latent projection are the output of some neural network. Why should these latent projection be physically consistent?
- Did you think of extending this approach to other modalities?

---

> ### Author Response · Authors · 2025-12-03
>
> Thank you for your consideration of our research. We appreciate your detailed feedback and insights emerging from your expertise in other forms of tomography. We have revised the paper to address the weaknesses you listed, rewriting the Methods section to be substantially more comprehensive and expanding the experimental section to include visualizations of reconstructed tomograms from the missing wedge synthesis experiment, as well as visualizations of generated tilts for the leave-one-out experiment. We also provide the results of ablation studies for all of the components you suggested, including the size of the latent dimension, amount of downsampling present in the encoder, the masking strategy used during training, and the adversarial loss.
>
> To address your first question, the latent projections are physically consistent because the convolutional architecture of the encoder maintains the physical geometry of the input tilt series. The convolution operation preserves spatial structure and, as each tilt is independently encoded, the latent projections retain the same physical relationship to each other possessed by the input tilts.
> For your second question, we are certainly interested in demonstrated that this approach will be robustly effective for other forms of limited-angle tomography. However, as our approach is a large pretrained model, we would likely need to a different or expanded training data set to provide an appropriate model for other modalities.

---

### Official Review · Reviewer_ZJhm · 2025-10-28

**Soundness:** 2
**Presentation:** 1
**Contribution:** 1
**Rating:** 2
**Confidence:** 4

**Summary:**

This paper proposes the Latent Back-projection Network (LBN), a generative model designed to address the missing wedge problem in cryo-electron tomography. LBN encodes tilt-series projections into a latent space, performs differentiable back-projection to reconstruct a 3D latent volume, and then synthesizes projections at uncollected angles. The model is trained in a self-supervised manner, learning to complete missing information under physical constraints. Compared with traditional reconstruction and IsoNet, LBN achieves lower errors and higher correlations on both real and simulated datasets. It can be easily integrated into existing Cryo-ET pipelines, improving reconstruction quality and downstream structural analysis.

**Strengths:**

The paper proposes a novel projection-level generative framework that integrates physical constraints through latent back-projection, offering a fresh and impactful solution to the missing wedge problem in cryo-electron tomography.

**Weaknesses:**

The paper suffers from serious issues in structure, clarity, and experimental rigor, making it feel more like a course project or technical report than a polished research contribution.

First, the writing and organization are weak: Figure 1 is placed above the title rather than as a teaser, and the Method section is confusing, with no formal definition of the latent volume and vague terms like “ResNet-like encoder.” There is also redundant text (e.g., using filter() instead of explicitly stating “ramp filter”), no mathematical formulation of the loss function, and critical training details are scattered in the Results section rather than logically grouped in Methods. The overall style is rough, the method description is unclear, and reproducibility is poor. I recommend splitting Figure 1 into two clearer figures, redrawing Fig 1c to be more concrete, and rewriting the Methods section with precise definitions and equations.

Second, the experimental section is weak: it only includes simple quantitative metrics and lacks qualitative visual results, which significantly undermines the credibility of the claims. There is also no ablation study to justify the design choices of the model. To strengthen the work, the authors should provide visualizations of reconstruction results, add pipeline and result figures for clarity, and conduct ablations to demonstrate the contribution of each model component.

In short, the work requires substantial polishing to improve clarity, insightfulness, and experimental support. It would be great if the authors can find an expert to guide the paper writing.

**Questions:**

N/A

---

> ### Author Response · Authors · 2025-12-03
>
> Thank you for your feedback, we appreciate the contribution of your domain expertise. We have revised the paper to substantially improve the writing and presentation of the Methods section. We have also expanded the presentation of our experimental results to include several ablation studies as well as qualitative results, presenting visualizations of generated tilts for the leave-one-out experiment and of the reconstructed tomograms for the missing wedge synthesis experiment.

---

### Official Review · Reviewer_KFqb · 2025-10-31

**Soundness:** 2
**Presentation:** 1
**Contribution:** 1
**Rating:** 2
**Confidence:** 4

**Summary:**

The paper addresses the critical issue of the missing wedge in cryo-electron tomography (Cryo-ET), which arises from limited tilt-series angular coverage and leads to anisotropic resolution loss and artifacts. The authors propose a Latent Back-projection Network (LBN) as a generative model to synthesize missing projections by learning a latent representation of the 3D volume. The method involves three stages: encoding 2D projections into a latent space, performing differentiable latent back-projection to form a 3D volume with a UNet for missing information recovery, and decoding to generate new projections at arbitrary angles. The model is trained on a large dataset of real Cryo-ET tilt series using a masked autoencoder approach, with evaluations on held-out real data and simulated datasets (e.g., SHREC 2021). Results are compared against conventional methods like weighted back-projection (WBP) and a state-of-the-art deep learning approach (IsoNet), using metrics such as MSE, Pearson correlation, and Fourier Shell Correlation (FSC).

**Strengths:**

​​Problem Relevance​​: The paper tackles a well-known and significant challenge in Cryo-ET—the missing wedge—which is a major bottleneck for high-resolution 3D imaging in structural biology.
​​Integration of Physical Constraints​​: The LBN architecture incorporates differentiable back-projection and re-projection operations, which embed physical principles of tomography (e.g., Fourier slice theorem).
​​Generative Capability​​: The model demonstrates the ability to synthesize projections at missing angles, which could potentially improve downstream tasks like tomographic reconstruction and subtomogram averaging. The masked training scheme and data augmentation (random rotations) help in generalizing to varied missing wedge scenarios.

**Weaknesses:**

First and foremost, the studied topic is relatively narrow for the ICLR community. Its broader impact outside the Cryo-ET field is limited, which may reduce the interest for a wider audience.
The core approach—based on autoencoders and back-projection—is relatively standard in image reconstruction literature. While the integration into a latent space is novel for Cryo-ET, the overall framework (e.g., ResNet/UNet components) lacks strong innovation compared to recent advances in generative models (e.g., diffusion models or transformers). This limits the perceived novelty and impact.
The method description in the main text is brief, with critical implementation details (e.g., network architectures, hyperparameters) relegated to the appendix. This makes it difficult to assess the depth of the contribution.
The comparisons are restricted to WBP and IsoNet, overlooking other recent deep learning methods. The results show marginal improvements (e.g., slight FSC gains in Figure 4), which may not justify the complexity of the model. Additionally, the evaluation on simulated data (SHREC) is not sufficiently rigorous—real-world applicability remains uncertain due to the idealized setup.
The "leave-1-out" experiment for real data (Table 1) may not fully capture the missing wedge problem, as it only omits single tilts rather than continuous angular ranges. The use of an auxiliary WGAN-like loss is mentioned but not adequately explained or ablated, raising questions about its necessity and effectiveness.

**Questions:**

​​Novelty Justification​​: Given that autoencoders and back-projection are well-established techniques, what specific aspects of LBN constitute a significant advancement over prior work?
​​Scalability and Generalization​​: The model is trained on a specific set of tilt series (CZI IDs 10000-10048). How does it perform on diverse, unseen data with varying resolutions or biological content? Are there limitations in handling large-scale volumes or high-noise conditions?
​​Quantification of Improvements​​: The FSC curves (Figure 4) show minimal resolution gains. What is the statistical significance of these results?
​​Ethical and Practical Implications​​: The generated projections are used to augment reconstructions. Are there risks of introducing hallucinations or biases in downstream biological interpretations? How does the model ensure physical plausibility beyond the training data?

---

> ### Author Response · Authors · 2025-12-03
>
> Thank you for reviewing our work. We have revised our paper to address your concerns regarding the lack of details in our Methods section, including providing explicit notation for the auxiliary loss.
>
> Our approach is less likely than alternative approaches to produce hallucinations as a result of the physical constraints provided by back-projection.

---

### Official Review · Reviewer_mMAw · 2025-10-31

**Soundness:** 2
**Presentation:** 2
**Contribution:** 2
**Rating:** 4
**Confidence:** 3

**Summary:**

In this paper, the authors propose a new method called LBN for cryo-electron tomography reconstruction. LBN encodes 2D projection images into a latent space, performs reconstruction of the missing wedge region within this space, and subsequently back-projects the results to the real space. The model demonstrates superior reconstruction performance compared to baseline methods on the CZI dataset.

**Strengths:**

- The idea of performing reconstruction in the latent space for cryo-electron tomography is quite interesting.
- The experiments demonstrate that the model achieves better performance than the baselines on the CZI dataset.

**Weaknesses:**

- The overall presentation could be improved. The paper would benefit from providing more background and a clearer formulation of the problem. In particular, the description of the proposed method is somewhat confusing and should be reorganized for better readability and logical flow.
- The paper lacks a systematic ablation study. It would be helpful to include analyses demonstrating how design choices affect performance. For example, the selection of the latent space dimension, the role and effectiveness of auxiliary losses, and the impact of each architectural component.
- There appear to be conceptually similar ideas in the 3D vision domain that perform reconstruction within a latent space. It would strengthen the paper to discuss and compare against these related works in the related-work section, highlighting what distinguishes this approach from existing latent-space reconstruction methods.

**Questions:**

- I am concerned about the model’s computational efficiency. It would be helpful to report the training speed and resource usage for both the baselines.
- The compared baselines appear somewhat outdated (from 2022). Please consider including comparisons with more recent state-of-the-art models.
- Additional ablation studies are encouraged to provide a more comprehensive analysis of the proposed method.

---

> ### Author Response · Authors · 2025-12-03
>
> Thank you for your consideration of your work. Following your feedback, we've rewritten the Methods section of the paper and incorporated several ablation studies into the experimental portion of the paper.
>
> Computational efficiency is not directly addressed in this paper because there is not a straightforward way to make a fair comparison between methods. All previously published deep-learning-based approaches to correcting for the missing wedge, like IsoNet, involve fitting the model to the data being corrected. That is to say, the computational cost of correcting for the missing wedge includes the cost of training as well as the cost of inference. Our approach is a general model, pretrained on a hundreds of tilt series. This training process takes considerably longer than the methods we compare to, however synthesizing missing tilts for a new tilt series is much faster than the training and inference sequence of IsoNet.

---

### Meta-Review · Area_Chair_ij2u · 2026-01-06

**Summary:**

The authors propose Latent Back-projection Network (LBN), an image-formation-aware masked autoencoder for cryo-electron tomography that aims to mitigate the missing-wedge artifact by learning to synthesize physically consistent tilt projections at angles that cannot be collected. The pipeline encodes each measured 2D tilt image into a compact latent, back-projects the set of latent projections into a single 3D latent volume via a differentiable latent back-projection module using the known tilt angles and a fixed ramp filter (from weighted back-projection), applies a lightweight 3D UNet to recover missing-wedge information in the latent volume, and then re-projects/decodes the refined volume to generate projections at arbitrary angles. Training is self-supervised via a masked-tilt scheme using MSE plus an auxiliary patch-wise critic loss (Wasserstein-inspired) as a perceptual similarity term.

**Reviewer Concerns:**

Most of the concerns remain, and they have been addressed only partially and in a very limited way. I believe the manuscript still requires substantial improvement in overall quality.

**Reviewer Scores:**

Overall, the rebuttals were not enough to address the reviewers' concerns.

---

### Decision · Program_Chairs · 2026-01-26

Reject